# CD36 and CD97 in Pancreatic Cancer versus Other Malignancies

**DOI:** 10.3390/ijms21165656

**Published:** 2020-08-06

**Authors:** Cristiana Tanase, Ancuta-Augustina Gheorghisan-Galateanu, Ionela Daniela Popescu, Simona Mihai, Elena Codrici, Radu Albulescu, Mihail Eugen Hinescu

**Affiliations:** 1Victor Babeș National Institute of Pathology, 99-101 Splaiul Independentei, 050096 Bucharest, Romania; daniela.popescu@ivb.ro (I.D.P.); simona.mihai21@gmail.com (S.M.); raducan.elena@gmail.com (E.C.); radu_a1@yahoo.com (R.A.); mhinescu@yahoo.com (M.E.H.); 2Faculty of Medicine, Titu Maiorescu University, 001863 Bucharest, Romania; 3Department of Cellular and Molecular Biology and Histology, Carol Davila University of Medicine and Pharmacy, 8 Eroilor Sanitari Str., 050474 Bucharest, Romania; agheorghisan.a@gmail.com; 4‘C.I. Parhon’ National Institute of Endocrinology, 001863 Bucharest, Romania; 5National Institute for Chemical Pharmaceutical R&D, 001863 Bucharest, Romania

**Keywords:** pancreatic cancer, CD36, CD97, cancer-associated fibroblasts, pancreatic ductal adenocarcinoma (PDAC), quiescence

## Abstract

Starting from the recent identification of CD36 and CD97 as a novel marker combination of fibroblast quiescence in lung during fibrosis, we aimed to survey the literature in search for facts about the separate (or concomitant) expression of clusters of differentiation CD36 and CD97 in either tumor- or pancreatic-cancer-associated cells. Here, we provide an account of the current knowledge on the diversity of the cellular functions of CD36 and CD97 and explore their potential (common) contributions to key cellular events in oncogenesis or metastasis development. Emphasis is placed on quiescence as an underexplored mechanism and/or potential target in therapy. Furthermore, we discuss intricate signaling mechanisms and networks involving CD36 and CD97 that may regulate different subpopulations of tumor-associated cells, such as cancer-associated fibroblasts, adipocyte-associated fibroblasts, tumor-associated macrophages, or neutrophils, during aggressive pancreatic cancer. The coexistence of quiescence and activated states in cancer-associated cell subtypes during pancreatic cancer should be better documented, in different histological forms. Remodeling of the local microenvironment may also change the balance between growth and dormant state. Taking advantage of the reported data in different other tissue types, we explore the possibility to induce quiescence (similar to that observed in normal cells), as a therapeutic option to delay the currently observed clinical outcome.

## 1. Introduction

Pancreatic cancer is a very lethal disease, with a five-year survival rate of 8% and very slow advances in its treatment [1,2]. Pancreatic ductal adenocarcinoma (PDAC) represents the seventh leading cause of cancer-related deaths worldwide. It is characterized by a high mortality rate, largely because of late diagnosis, early metastasis, and limited reaction to chemotherapy or radiotherapy. Unfortunately, most of the patients with pancreatic cancer fail to develop important symptoms before reaching the advanced stage of the disease. Moreover, the CA19-9 antigen test, currently in use, is not sufficient to diagnose pancreatic cancer with high sensitivity and specificity [3]. According to GLOBOCAN data, 458,918 new cases of pancreatic cancer and 432,242 new deaths were recorded in 2018. By 2040, 355,317 new cases are estimated to occur. The five-year survival rate still stands at 9%. By 2030, pancreatic cancer is projected to be the third cause of cancer-related death. The main therapeutic approach for this malignancy is surgical resection, followed by adjuvant chemotherapy that includes 5-fluorouracil/leucovorin with irinotecan and oxaliplatin (FOLFIRINOX) and gemcitabine/nab-paclitaxel. However, the development of chemoresistance among PDAC patients leads to poor clinical outcomes. Studies focused on this disease suggested that PDAC chemoresistance is a result of the interaction between pancreatic cancer cells, cancer stem cells, and the tumor microenvironment [4,5,6,7,8,9].

Several risk factors have been associated with PDAC, including tobacco smoking, diabetes mellitus, obesity, dietary factors, alcohol abuse, age, ethnicity, family history and genetic factors, *Helicobacter pylori* infection, belonging to non-0 blood group, and chronic pancreatitis [6,10,11,12]. The inflammation and immunosuppression caused by microbiome changes are other factors involved in the development of PDAC, and they are able to affect the metabolism of chemotherapy [13].

Besides PDAC, which represents the most fatal tumoral disease of the pancreas (covering about 90% of the total cases), several other cancers are present in the pancreatic environment. For PDAC, there is the need of molecular subtyping, thus advancing the need of a framework of molecular taxonomy. Several ductal lesions are considered tumor precursors, and a standard was adopted recently for the classification of pancreatic intraepithelial neoplasia (panIN). Molecular investigation demonstrated that PanIN-2 and -3 represent distinct steps toward invasive carcinoma. Several advances were made in further immunocytochemical and molecular characterization of other pancreatic neoplasms—mucinous noncystic carcinoma, undifferentiated mucinous cystic neoplasm, intraductal papillary mucinous neoplasm, medullary carcinoma, and other rare tumors of the pancreas [14] (see Figure 1).

In this review, we examine the recent literature, in order to explore the hypothesis that the induction of quiescence in pancreatic cancer, either in tumor cells or in tumor-associated cells, could be a putative valid therapeutic strategy. The recent recognition (in other types of tissues) of CD36 and CD97 as markers of quiescence compelled us to examine if our hypothesis could be supported by experimental facts available in the literature.

## 2. CD36 in Pancreatic Cancer vs. CD36 in Normal Tissues: Where Do We Stand?

### 2.1. CD36 in Normal Tissues

CD36, a scavenger receptor class B type 2 (SR-B2), is a transmembrane glycoprotein that is expressed on the cell surface in multiple cell types, including dendritic cells, microvascular endothelial cells (MVECs), retinal epithelial cells, platelets, monocytes/macrophages, erythrocytes, adipocytes, microglial cells, podocytes, skeletal muscle cells, mammary epithelial cells, taste receptor cells, hepatocytes, Kupffer cells, enterocytes, and serous ovarian epithelial cells. CD36 molecule was examined during several diseases, including cancer, where it seems to support development of metastasis.

In the pancreas, CD36 was found in the plasma membrane, as well as intracellularly and co-localized with insulin granules. CD36 activity appears important for the uptake of fatty acids (FAs) into β-cells, as well as for mediating their modulatory effects on insulin secretion [17]. In a comparative study, exploring pancreatic cancer versus normal pancreatic tissue, CD36 was found to be significantly lower in cancer than in corresponding non-tumor normal tissues [18].

Exposure to the ligand determinates CD36 to dimerize. In some membrane microdomains, such as caveolae, a special type of lipid rafts that are rich in proteins and lipids, CD36 can copolymerize with caveolin-1, suggesting the participation of the two molecules together in the activation of the signaling pathways [19].

Furthermore, CD36 may associate with other transmembrane proteins, such as integrins (*β*1, *β*2, and *β*5) and four-transmembrane proteins named tetraspanins (CD9 and CD81), which jointly mediate ligand binding and signal transduction [20]. CD36 intracellular domains, one single short cytoplasmic tail at each terminal (N and C), associate with members of the Src family of tyrosine kinases. A molecular interaction is most probably mediated by lipids in the context of lipid rafts [21]. Having a wide distribution in membrane-bound and cytoplasm organelles, such as mitochondria, endosomes, and endoplasmic reticulum (ER), CD36 promotes FA oxidation by itself or in cooperation with carnitine palmitoyltransferase-1 (CPT1) in mitochondria, along with maturation and ubiquitylation-mediated inactivation of CD36 in the ER [22]. CD36 can be transported to organelles and cell membrane by intracellular and extracellular vesicles.

CD36 transport to the cell membrane can be facilitated by several physiological stimuli, the most potent of which are (a) insulin—by activating the phosphatidylinositol 3-kinase (PI3K)/AKT signaling axis; (b) muscle contraction—by activating adenosine 5ʹ monophosphate-activated protein kinase (AMPK); and (c) inflammation [23,24].

CD36 is also known as a fatty acid translocase (FAT), because it imports long-chain fatty acids (LCFAs) in cells. The miscellaneous lipid and protein related ligands of CD36 contribute to its versatile functionality. Regarding the lipid-related ligands category that binds CD36, we can include LCFAs, anionic phospholipids, and oxidized lipids such as low- and high-density lipoprotein (ox-LDL and ox-HDL) and oxidized phospholipids (ox-PLs). CD36 has been involved in FAs transfer to cytosolic FA binding protein (FABPc) that mediates its passage to mitochondria. The lipid binding process described above has a major role on the cell’s energy metabolism [25].

Furthermore, CD36 has protein-related ligands with variated functions, among which we list the following: amyloid proteins, thrombospondins (TSP) 1 and 2, advanced glycation end products (AGEs), and advanced oxidation protein products (AOPPs). CD36 expresses on MVECs a specific domain for TSP-1 ligand (the CLESH domain) that generates Src-family pathway activation and promotes endothelial cells apoptosis [26,27].

Expression of CD36 on immune system cells, like dendritic cells and macrophages, promotes recognition and binding of apoptotic cells, respectively, β -amyloid peptides, AGEs, and AOPPs [28,29,30,31,32].

### 2.2. CD36 Promotes Tumor Metastasis in Pancreatic Cancer

Numerous studies confirmed the relationship between CD36 and metastasis. Investigation of CD36 in cancer revealed the role of CD36 in tumor metabolism, as well as in tumor immuno-editing, anti-angiogenic processes, metastasis, or therapy resistance. By associating with different ligands, CD36 is involved in cancer development [33].

Metastasis requires cellular changes related to cell-to-cell and cell-to-matrix adhesion, immune surveillance, activation of growth and survival signaling pathways, and epigenetic modifications. To be effective, these changes must occur in a time-dependent manner, modifying the cell phenotype for survival in new microenvironments [34].

CD36 contributes to the progression and metastatic potential of cancer by several mechanisms, such as activation of cancer stem cells, epithelial-to-mesenchymal transition, and chemoresistance [35,36].

CD36 can be a prognostic marker for different cancers, most often of epithelial origins, such as breast cancer, ovary cancer, prostate cancer, or hepatocellular carcinoma. It was proposed as an “early prognostic marker” for metastasis in gastric, ovary, and breast cancer; oral squamous cell carcinoma, esophageal squamous cell carcinoma, or hepatocellular carcinoma; or as “an unfavorable prognostic factor” in lung, bladder, breast, and PDAC [36,37,38]. In PDAC, the decreased expression of CD36 is associated with large tumor size and reduced survival rate, and less associated with TNM staging [18].

Out of a study on more than 2500 cases of different cancers came the confirmation of a role of CD36 in metastasis by investigating genes implicated in metabolic reconnection to aerobic glycolysis and fatty acids synthesis in metastatic vs. primary tumors. It was found that the CD36 gene appeared to be frequently amplified in metastatic tumors. Moreover, survival rates were reduced in the high-copy number group, as compared to the low copy group [39].

CD36 is expressed in tumor tissues, not only by tumor cells, but also by stromal, immune cells, and MVECs, and depends on tumor stage and cell type. Experimental data suggest that CD36 has a minor role in the initiation of the primary tumor, but its implication is significant in starting metastasis process. In tumor microvessels that support tumor development and metastasis, expression is generally downregulated. In the tumor stroma, CD36 expression is also deficient; the lower the CD36 level in the stroma, the more aggressive the tumor [40,41,42,43].

It has been shown that, in some forms of cancer, such as colon, breast, and ovarian, a low-CD36 expression in the primary tumor is associated with higher metastasis grade and poor prognosis. It was demonstrated that CD36 has significantly lower expression in pancreatic cancer cells’ lines and tumor tissues [18]. It was suggested that low expression of CD36 might reduce tumor cell adhesion to the extracellular matrix, followed by an increase of cell mobility due to decreased ability of CD36 to bind collagen [44].

Experimental data suggest that CD36 is involved only in metastasis initiation and proliferation of metastatic cells. The uncontrolled division of tumor cells requires high energy. The cellular metabolic pathway that provides the most energy is β-oxidation of FAs. Thus, a large number of FAs’ molecules via CD36 supports cancer cell proliferation [45]. It can be concluded that lipid metabolism is involved in the survival of migrated tumor cells in a new microenvironment, and increasing the expression of CD36 in these cells may be a marker that supports their proliferation.

There is evidence suggesting that, in some forms of cancer, such as hepatocellular carcinoma, fatty acid uptake through CD36 may promote cancer cell metastasis and distant proliferation [46].

The tumor metastasis-initiating cells derive from the primary tumor and contribute to seeding metastases in other organs [47]. Involvement of CD36 in the metastatic process is related to the three components of any tumor niche: the tumor cells, the stromal cells, and the endothelial cells. In cancer studies, CD36 is investigated mostly in relationship with thrombospondins (TSPs) and, to a lesser extent, with transforming growth factor-β (TGF-ß).

CD36 protein expression can be modified in metastasis via epigenetic modifications and post-transcriptional interference of non-coding RNA, as was recently suggested. As such, in certain cell types, regulation of CD36 expression involved DNA methylation or histone tails or miRNA interference [48].

CD36-induced potential of metastasis was reported to depend on lipid metabolism in cancer cells. CD36 mediates the FA uptake, key nutrients for tumor metabolism. In the case of gastric cancer, uptake of palmitic acid, mediated by CD36, was demonstrated to activate AKT phosphorylation and inhibit the degradation of glycogen synthase kinase 3β (GSK-3)/β-catenin, thus promoting metastasis [17].

Tumor-associated adipocytes provide enough fatty acids to the tumor cells and support proliferation and metastasis. For instance, gastric cancer often metastasizes in the greater omentum, rich in adipocytes. Adipocytes induce CD36 expression in metastatic ovarian tumors [40]. In the oxidation of fatty acids, the rate-limiting step is their transport to mitochondria, a process having as key enzymes CPT1 and CPT2. In human skeletal muscle cells, CD36 on mitochondria is able to bind CPT1, and upregulation of mitochondrial CD36 correlates with increased oxidation of fatty acids. In the case of oral squamous cell carcinoma cells, CD36 blockade generates intense intracellular lipid accumulation, leading to lipotoxic cell death and vicious metastasis [40,45].

Tumor-associated neutrophils are studied to a lesser extent, in pancreatic cancer; Zhang et al. found that in neuroendocrine tumors their presence predicts a poor survival [49].

Sano et al. documented a shift in immune-inflammatory microenvironment in a mouse model of PDAC, supporting the idea that tumor-stromal interaction could be a therapeutic target [50].

Apparently, even in pancreatic cancers, tumor associated neutrophils contribute to maintenance of a “permissive tumor microenvironment” [51].

The epithelial–mesenchymal transition promotes cancer-cell metastasis; a large number of studies regarding epithelial–mesenchymal transition (EMT) in all cancers were published [52,53]. Existing data suggest that EMT has a significant role in the development of the tumor budding, which contains one to five highly aggressive non-proliferating neoplastic cells at the infiltrative front of the tumor. Tumor budding becomes responsible for the invasion of the peritumoral connective tissue, and the infiltration of the lymphatic and blood vessels [54]. Thus, tumor budding is considered as an indicator of cancer invasiveness, including PDAC. Many processes are involved in EMT, including the reorganization of cell-surface and cytoskeletal proteins, low expression of E-cadherin, the activation of the zinc finger transcription factors that repress genes responsible for the epithelial phenotype (e.g., ZEB1, SNAIL, Slugand, and Twist), acquisition of mesenchymal markers (e.g., N-cadherin, Vimentin, and Fibronectin), increased production of extracellular matrix components, and changes in the expression of specific microRNAs [55]. Thus, epithelial cells lose their polarity and adhesion, acquire migratory and invasive capabilities, and become resistant to apoptosis.

In PDAC, tumor-budding cells and adjacent stromal cells showed increased levels of the E-cadherin repressors ZEB1, ZEB2, and SNAIL1. Moreover, tumor-budding cells lose expression for membrane adhesion molecule E-cadherin and β-catenin, without detectable nuclear β-catenin, and favorize tumor budding detachment from the primary tumor [56]. High expression of ZEB1 in tumor-budding and stromal cells was correlated with high peritumoral invasion. It was demonstrated that ZEB1- and ZEB2-positive stromal cells are cancer-associated fibroblasts, and it was suggested the existence of many subtypes of stromal cells phenotypically distinct [57]. A high level of miR-21 and a low level of miR-200c expression were associated with pancreatic cancers [54].

CD36 attenuates angiogenesis by binding to TSP-1 and thereby inducing apoptosis or blocking the vascular endothelial growth factor receptor 2 pathway in tumor microvascular endothelial cells [58].

Despite its anti-angiogenic action, TSP-1 might have a contrary effect suggested by promotion of metastatic behavior (increased production of cancerous emboli and enhanced adhesion of cancer cells) [59]. Activated TGFβ, released by a RFK/WXXW–TSP-1 interaction, is involved in tumor cells’ expansion mechanism by enhancing matrix production and altering expression of integrins, and it promotes upregulation of plasminogen activator, its receptor, and plasminogen activator inhibitor-1 [60,61].

In some cancers, such as breast, prostate, gastric, and lung, a partial link has been observed between the metastatic effect and the concentration of TGFβ at the tumor site.

The opportunity of using TSP-1 as an anti-angiogenic therapeutic target in cancer treatment is overshadowed by the divergent effect of TSP-1 (pro-angiogenic, on the one hand, by the TGFβ release, and the anti-angiogenic effect described above, on the other hand).

### 2.3. CD36—A Mediator of the Engulfment of Pancreatic Tumor Microvesicles

In the blood of patients with metastatic cancer circulate extracellular vesicles named tumor microvesicles, which could play a major role in intercellular communications and have been suggested as an early tumor-detection marker [62]. These vesicles act as carriers for various RNA species. They are different from exosomes in what concerns biogenesis, composition, and biological functions and participate in the progression of several types of cancer (prostate, colorectal, and pancreatic cancer) [63]. They transfer bioactive cargoes to both adjacent and distant sites, and they orchestrate carcinogenesis and malignant progression [64]. Microvesicles play an important role in mediating immune system response in metastasis progression and can also influence tumor cells phenotype (transition from a weak to a highly invasive phenotype) through the transfer of the epidermal growth factor receptor variant EGFRvIII [65].

This mechanism has been broadly researched in **metastatic pancreatic tumors**, following the trajectories taken by pancreatic tumor microvesicles in the liver microcirculation, the major site of pancreatic cancer metastasis [66]. A fraction of the tumor microvesicles present in the liver microcirculation have the ability to cross liver sinusoids endothelial layer via CD36 receptor and relocate in perivascular Ly6C2 macrophages for at most two weeks. Ly6C2 macrophages, which are different from Kupffer cells, can originate from recruited Ly6C2 patrolling monocytes or from extravasated inflammatory monocytes [67]. Thus, the microvesicles are increasingly integrated into CD36-induced premetastatic cell clusters and enhance development of liver metastasis. The persistent infiltration of perivascular macrophages with tumor microvesicles was associated with an augmented survival of extravasated tumor cells [66]. In vivo mice liver macrophages and in vitro myeloid immune cells studies confirmed the important role of CD36 in tumor microvesicles mediation [66,67]. Intravesicular cargo of microvesicles transferred to immune cells via CD36 can persist in these cells for extended time periods, and, thus, CD36 could potentially support the long-term reprogramming of cellular phenotypes relevant for tumor metastasis [66].

### 2.4. CD36 Can Regulate Chemoresistance in Pancreatic Cancer

In medical practice, cancer raises two major problems: early diagnosis and resistance to therapy. A possible correlation between CD36 expression and chemotherapy resistance has been studied in patients with PDAC treated with Gemcitabine. Many of them exhibited resistance to treatment after a short time, with consequently poor prognosis. The Gemcitabine resistance and the poor outcome in these patients were related to a CD36 strong expression correlated with a significant microinvasion to the venous system [38]. Moreover, patients expressing high levels of CD36 showed a worse prognosis in survival statistics. The unfavorable outcome could be explained by a more severe clinical–pathological picture, with microinvasions of the venous system also significantly correlated to CD36 expression [38].

It has been suggested that CD36 expression influences gemcitabine resistance by regulating anti-apoptosis proteins such as B-cell lymphoma-2 (Bcl-2), B-cell lymphoma extra-large (Bcl-xL), and myeloid cell leukemia-1 (Mcl-1). The results indicated that CD36 could enhance anti-apoptosis protein expression, which contributed to gemcitabine resistance by protecting cancer cells from drug-induced cell death. Thus, high CD36 expression is an unfavorable prognostic factor in PDAC [38,68].

Alone or with gemcitabine, quercetin administered orally in the diet has been reported to inhibit pancreatic cancer. Although the mechanisms have not been elucidated and the results are divergent, CD36 was suggested as a possible target for quercetin because this flavonoid promotes the cell adhesion, regulates the thrombospondin-1 activity, and increases FAs uptake and oxidation by activating glutathione transferases [69].

Based on the data presented, one may consider that there are just a few studies on CD36 expression in normal pancreatic tissue and cancer.

## 3. CD97 During Pancreatic Cancer vs. CD97 in Normal Tissue

### 3.1. Distribution and Functions of CD 97

CD97 is a member of the seven-trans-membrane subfamily of the class B G protein-coupled receptor (GPCR) group of the epidermal growth factor (EGF) and is present on the surface of lymphocytes, monocytes, macrophages, dendritic cells, granulocytes, and smooth muscle, being a dimeric glycoprotein with a 75–90 kDa intracellular domain and a 28 kDa extracellular domain [70,71]. CD97 is expressed on T-cells, but rarely on B cells. CD97 is intended to be an important part in cell adhesion, migration, and regulation of intercellular junctions. Various studies reported CD97 as being induced or upregulated, and/or biochemically modified in various malignancies, including of those of the thyroid, stomach, colon, prostate, pancreas, and brain, as compared to the normal tissues in question [72,73,74,75,76,77,78,79]. Moreover, CD97 was related to an invasive phenotype, correlated with tumor grade, invasion of the lymph node, metastatic spread, and overall prognosis [73,76,78,80]. The association of CD97 with human cancers represents an emergent subject of research in recent years.

In normal tissues, excessive CD97 expression was found only in macrophages and dendritic cells, excepting neuroglia, and some T and B cells [81].

Based on its unique expression pattern and structure, CD97 might play key roles in cellular adhesion, through connections with other proteins of the cell surface and of the extracellular matrix. In humans, three ligands for CD97 were identified. The first is CD55, also known as decay-accelerating factor, which interacts with EGF domains, a negative regulator of the complement cascade. The second is glycosaminoglycan chondroitin sulfate, which binds specifically to the large isoform(s) of CD97 and affects cell attachment [82,83]. The third category of ligands includes integrins, such as α5β1 and αvβ3, which bind a (arginine-glycine-aspartic acid) RGD motif in the stalk region of the CD97 α-chain 19 [84,85,86].

It was revealed that, during chronic inflammatory processes, soluble forms of CD97 protein were expressed in various body fluids. Based on the fact that chronic inflammation accompanies all the malignancies, it was observed that certain types of cancer (e.g., gastric cancer) display an overexpression of CD97 in comparison to normal individuals [87,88,89,90,91,92].

### 3.2. Expression of CD97 in Pancreatic Cancer

Discrimination between pancreatitis (PT) and PDAC represents a significant problem concerning the perioperative assessment of pancreatic tissue frozen sections.

The dynamics of the expression of CD97, CD95, and Fas-L in pancreatic tissues, using immunohistochemical evaluation, could be a potential diagnostic marker for the separation of inflammatory syndromes versus malignant neoplasms in the perioperative evaluation of pancreatic cryo-cut sections. In human pancreatic ductal adenocarcinomas, it could be a possible marker for dedifferentiation, invasiveness, or aggressive activity. This study revealed that CD97 expression was strong only in PT; its expression is weak in poorly differentiated PDAC. Since CD97 was not expressed by normal pancreatic tissue, it was concluded that CD97 could be considered a useful marker for PT and undifferentiated carcinomas. Thereby, in cryo-cut sections, CD97, CD95, and Fas-L can be used as additional markers to differentiate between PT and well-differentiated PDAC [90].

Even though CD97 and CD55 were considered to be defensive mechanisms in relation to the complement immune system, the presence of a small population that significantly expresses CD97 and CD55 appears to be correlated with a poor prognosis in some malignancies. Several studies have shown that both CD97 and CD55 play important roles in dedifferentiation, migration, invasiveness, and metastasis of tumors [71,93]. The association between expression of CD97 and CD55 in pancreatic cancer was not sufficiently investigated yet.

He et al. observed by immunohistochemistry analysis that, as the expression of CD97 and CD55 increases, a deterioration in cancer prognosis occurs, closely associated with lymph node involvement, metastasis, and vascular invasion [75].

Furthermore, Vogl et al. [94] revealed that the levels of CD97, CD274, and CD276 assessed by ELISA could serve as readily measurable prognostic or predictive markers in patients with advanced disease or metastatic breast, colon, or pancreatic cancer, being at baseline before cytotoxic treatment, and during the course of the chemotherapy, as well.

Consequently, it has been demonstrated that CD97 levels expressed considerable variability during the course of chemotherapy [94]. No correlations were found between CD97 expression, clinical infection, or C-Reactive Protein level, being hypothesized that infection could activate CD97 through upregulation of its ligand, CD55. No significant correlation was found between CD97 expression and tumor response, as well, in colorectal or pancreatic cancer.

However, CD97 levels are prognostic for overall survival, and thus they predefine the disease’s aggressiveness. Chemotherapy also affects certain cancer cell clusters which express the membranous form of the molecule CD97 [94].

In spite of the fact that CD97 was not or was vaguely expressed in the corresponding normal tissues of various analyzed tumors, its expression was positive in pancreatic ducts, the origin of progenitor cells, and for the majority of pancreatic adenocarcinomas.

Aust et al. have demonstrated the presence of CD97 in gastric, pancreatic, and esophageal tumors, revealing the implication of CD97 in the invasion of tumor cells, possibly as a differentiation-dependent or adhesion molecule. It should be noted that these studies used RT-PCR and flow cytometry for determination of CD 97 and EMR2 expression at the messenger RNA and protein levels, followed by immunohistochemical methods [89].

Given all these aspects, it remains questionable whether CD97 acts in an analogous manner in pancreatic cancer or whether CD97 exerts potential roles in the differentiation processes involving pancreatic progenitor cells.

## 4. Why Examine Concomitant Expression of CD36 and CD97s? (Why Bother with CD36 and CD97 in Pancreatic Cancer?)

Advances in surface proteome analysis and CD markers’ discovery might offer valuable insights concerning the metastatic niche in pancreatic cancer. There are few studies in literature that tackle the surface proteome profiling in different organs, to key cellular events in oncogenesis or metastasis development. Facts with reference to a potential common contribution of CD36 and CD97 were revealed firstly by Heinzelmann et al. [95]. They studied expression changes in different CD markers profile in lung fibrosis, with a major emphasis on specific phenotypes during fibroblast–myofibroblast activation by TGFβ, known to express αSMA (α-smooth muscle actin). Thus, it is accomplished the phenotype switch into a highly proliferative and migrating one, with impacts on extracellular matrix (ECM)-producing cell types in the lung. Moreover, it was reported the presence of a minor population exhibiting a strong expression of both CD36 and CD97 in remodeled areas of idiopathic pulmonary fibrosis tissue, but αSMA-negative (by immunofluorescent staining), suggesting that they were not activated fibroblasts, but more likely being considered as indicator of a quiescent, non-proliferative fibroblast background. It was also observed that CD36- and CD97-positive population decreased upon TGFβ stimulation and was part of a senescent population, as well, being significantly increased in high passages. The simultaneous presence of quiescent and activated fibroblasts could be mirrored by dynamic changes in surface markers; thus, different fibroblast phenotypes are characterized by various combinations of CD expression [95].

Taking into consideration that these two markers were separately described in different studies [35,53,71,79] as conferring an invasive phenotype, silencing both their expression could affect the physiopathology of the disease.

It is hypothesized that cellular senescence could negatively impact cancer development, by shaping its surroundings toward a pro-carcinogenic microenvironment, with the accumulation of mutations over time [96]. Since fibroblasts are thought to be key players in the tumor microenvironment, deciphering their cancer-specific features by their different expression in CD markers on activated fibroblasts could open new avenues in the evolving concept of cell identity [97].

Due to the unique versatility of cancer-associated fibroblasts (CAFs) (which are particularly heterogeneous and highly plastic), the distinction between a CAF and a normal one within the tumor microenvironment is considered to be functional, is and less defined by specific biomarkers expression or other features [98].

The lack of specific markers that can be used in order to identify CAF populations derives from their own heterogeneity, which reflects a similar situation encountered in cancer stem cells. Both populations express distinct markers which show great variation during the disease and are defined more with reference to a specific cell state, rather than to a distinct cell type. This may be hypothesized that CAFs could behave as a dynamic state of fibroblasts [98].

In light of these findings, the question whether CD36 and CD97 would complement each other remains open. Further studies are needed to gain insight into the concomitant impact of CD36 and CD97 on modulating fibroblasts phenotypes under different conditions [95], thus offering potential innovative therapeutic strategies to inactivate CAF and prevent aberrant tumor-stroma crosstalk in pancreatic cancer.

## 5. Heterogeneity of Pancreatic-Cancer-Associated Fibroblasts

### 5.1. Tumor Microenvironment

The unique tumor microenvironment emerges as a result of complex interactions between tumor cells, a wide range of stromal cell types, which refer to the non-malignant cells in the tumor microenvironment (mostly fibroblasts, endothelial cells, and immune cells—T cells, neutrophils, and macrophages), blood vessels, inflammatory cells, and a wide diversity of associated tissue cells. The dense extracellular matrix is present in various tumors, acting as a barrier to drug delivery, or as a nutrient supply for tumors [99,100]. A specific characteristic of pancreatic cancer resides in its tumor composition, being represented in a percentage of 90% of stroma cells, and only a minority of them are cancer cells [100,101].

### 5.2. Normal Fibroblast

Among the most studied cells, fibroblasts remain “enigmatic and mysterious”, particularly due to the lack of a unique/specific marker; hence, they are characterized based on their morphology, tissue position, and lack of lineage markers specific for epithelial, endothelial cells, and leukocytes.

Fibroblasts are usually quiescent/resting cells and are reversibly activated in response to tissue injury, being involved in synthesizing extracellular matrix (ECM) proteins, the production of cytokines/chemokines, the enrolling immune cells, and modifying tissue architecture, and thus participating in wound healing process [102,103].

Markers for fibroblast subtypes have been identified, including Vimentin and platelet-derived growth factor receptor-α (PDGFRα), but together with other standards, like cell site or cellular shape, α-smooth muscle actin (αSMA), and fibroblast activation protein (FAP), assigning important roles in bone and fat homeostasis [104,105,106].

In pancreatic tissue was observed a distinctive type of fibroblast—quiescent pancreatic stellate cells (PSCs), which accumulate lipid droplets of vitamin A [107]. In their activated form, PSCs become proliferative and attain an expansive secretome, expressing αSMA marker, and losing the lipid droplets [108]. It was observed that the equilibrium between quiescence and activation cells is mediated by the vitamin D receptor; in its absence, spontaneous pancreas fibrosis is generated. Moreover, other studies [109,110,111] have assigned an important role in metabolic homeostasis to these stellate cells, indicating that fibroblasts are no longer simply producers of ECM, but are involved in a complex networking with different other cell types, playing important roles in both normal tissue homeostasis and repair [106].

### 5.3. Cancer-Associated Fibroblasts (CAFs)

It was revealed that fibroblasts become irreversibly activated within tumors, being epigenetically modified, and represent a key player of the tumor microenvironment, having a diversity of functions, including matrix production and remodeling, extensive reciprocal signaling interactions with cancer cells, and crosstalk with infiltrating leukocytes, both metabolically activated and proliferative [106].

In clinical practice, when analyzing a tissue biopsy, CAFs cells are identified by using both exclusion and inclusion criteria. Thus, exclusion criteria are negative staining for epithelial, endothelial, and leukocyte markers; an elongated morphology; and the absence of mutations found within cancer cells. Usually, these exclusion criteria are associated with positivity for a mesenchymal marker, like vimentin (which does not exclude other mesenchymal lineages, such as pericytes or adipocytes) [106]. In conclusion, is difficult to define CAFs cells, partially because of the lack of precision in defining its specific markers.

Based on the results of several experimental studies, in which it was observed that such cells exhibit distinctive characteristics compared to normal fibroblasts, it was highlighted that CAFs are extremely heterogeneous and highly plastic than was previously believed [112,113].

CAFs play an essential role in the multistep processes of promoting tumor initiation, progression, invasion, and metastasis, having a dual action: first as a barrier to immune surveillance and drug delivery, and second by secreting survival factors [100,114,115,116,117].

Multiple mechanisms of CAFs’ activation, following cancer cells–fibroblasts contact, have been proposed, and they are summarized in Table 1.

Within the tumor microenvironment, CAFs could be activated by different signals and stimuli from other cells, including macrophages inducing granulin, which promotes the activation of a fibrotic environment [126].

It was revealed that CAFs could be initiated from different cell types: activated tissue fibroblasts, transdifferentiated epithelial cells, or pericytes; and mesenchymal progenitor cells recruited into the tumor, stem cells [127,128,129], or pluripotent-adipose-tissue-derived stromal cells [102,130].

DeFilippis et al. found that the transmembrane receptor CD36, normally expressed in all stromal cells, including disease-free fibroblasts, is drastically decreased in CAF population [131]. They noticed that fibroblasts with a low expression of CD36 produced increased amounts of collagens and fibronectin, compared to fibroblasts with high expression of CD36. Similarly, low expression of CD36 in different cell types generates abnormalities, as observed in endothelial cells, which have shown an amplified angiogenesis in preadipocytes, which were incapable of differentiating into adipocytes, and in immune cells, which exert a pro-tumorigenic state (cells in M2), instead of an anti-tumorigenic state (cells in M2). Thus, a decreased expression of CD36 in CAFs cells was associated with disease progression [131,132].

### 5.4. Cancer-Associated Fibroblasts (CAFs) in PDAC

In the microenvironment of pancreatic cancer, CAFs represent the most abundant and heterogeneous population of stromal cells, originating both from the tumor itself, and from its surroundings, as well, having as mainly precursors the pancreatic stellate cells (PSCs).

Remarkably, in different types of cancers were noticed different CAFs phenotypes, thus generating an extraordinary heterogeneity [127,133]. Based on their complex roles, CAFs could be divided into five categories: tumor-promoting (pCAF), tumor-retarding (rCAF), secretory (sCAF), inflammatory (iCAF), and myofibroblast (myoCAF) [102,129]. In PDAC, the most common and intensively analyzed CAF subtypes are myoCAFs and iCAFs [102,106,134].

A combination of cellular markers is used to identify typical stromal fibroblasts, such as αSMA, chondroitin sulfate proteoglycan (NG2), FAP, fibroblasts specific protein-1 (FSP-1), and PDGFR [127]. While αSMA and FAP are known to be characteristic to myoCAFs, the specific marker for iCAFs is represented by the increased expression of inflammatory cytokine IL-6 [102,135].

The shift into iCAFs or myoCAFs depends on cellular signaling that implies TGFβ and IL-1 [136]. For example, a recent analysis of squamous PDAC has underlined a potential role of p63 in the initiation of the IL-1α-driven conversion of PSCs into iCAFs [102,137].

The involvement of PSCs in the synthesis process of collagens, fibronectin, laminin, and other ECM components, is marked once PSCs are activated, as a consequence of the transdifferentiation into myofibroblasts, thus contributing to the desmoplastic reaction that worsens the outcome of anticancer therapies, being recognized as one of the main hallmarks of PDAC [138]. In addition, the presence of CAFs in distant metastases led to the hypothesis that CAFs play major roles in cancer spread [102,139].

Moreover, the CAFs’ distribution in PDAC is different according with the distance from cancer cells. Thus, CAFs expressing high levels of αSMA (myoCAF phenotype) are in proximity to neoplastic cells, while CAFs expressing higher levels of IL-6 (iCAFs phenotype) are more distantly distributed inside the tumor [134]. The two distinct phenotypes could be a result of TGFβ-mediated suppression of the IL-1 receptor, further implied in NF-κB signaling and subsequent IL-6 expression [106,136].

CAF heterogeneity raised new questions, such as whether CAF subtypes might interconvert with each other or not. CAFs isolated from a PDAC mouse model can be reversible from the αSMA-high and IL-6-producing states through modification of TGFβ and IL-1 tumor-secreted ligands, claiming for remarkable plasticity in fibroblast phenotypes [106,136].

### 5.5. Secretome

CAFs possess an active and dynamic secretome, releasing a multitude of soluble factors, and induce numerous phenotypes in adjacent cells [132,140]. There are various molecular mediators which exert their actions at different levels, and they can be categorized according to factors that are highlighted in Table 2.

### 5.6. Metabolism

The metabolites exchange between CAFs and neoplastic cells is an important path by which stromal fibroblasts are connected with cancer cells [155,156,157,158]. Autophagy in stromal fibroblasts could produce alanine, which is later used by PDAC cells, especially PSCs, and incorporated in the tricarboxylic acid cycle [155,159].

Recent studies have illustrated that glutaminase expression was increased in CAFs compared to pancreatic tumoral cells, being the reason for which CAFs were more sensitive to glutamine removal than tumoral cells [100,160]. It was also observed that CAFs secrete greater levels of glutamate and glutamine in culture, sustaining the growth of pancreatic tumoral cells, compared to normal PSCs [161]. Nevertheless, future studies are needed to elucidate the role of glutamine metabolism in fibroblasts [100].

A study on patients with PDAC, conducted by Shan et al., revealed that the loss of stromal caveolin-1 is linked with poor clinical outcomes, affecting tumor development through metabolism transformation [162]. Moreover, downregulation of caveolin-1 and up-regulation of monocarboxylate transporter 4 was detected in CAFs and tumor cells, transferring lactate into neoplastic cells to growth ATP formation [160,163].

Until now, few things are known regarding heterogeneity in CAF metabolism and how it is associated with different CAF subtypes; nevertheless, it is likely that CAFs display a wide range of metabolic profiles, depending on the different metabolites’ availability. Deciphering the mechanisms involved in heterogenous cancer cell–CAF metabolic coupling will open novel therapeutic strategies in PDAC.

### 5.7. Challenges to Studying Metabolic Interactions in the Tumor Microenvironment

The application of common in vitro culture systems in metabolic studies comes with many disadvantages, due to the tumor heterogeneity and stromal content. Thus, new approaches are needed; one of them could be the 3D cell culture models/organoid cultures that mimic aspects presented in neoplastic and stromal cells [164,165,166,167].

Moreover, tissue slice cultures can be used in metabolic studies, mainly because they still maintain the metabolic features and cellular diversity of tumors [100,168].

In conclusion, metabolic studies have to take advantage of using organoid systems due to their enhanced mimicking of tumor heterogeneity and nutrient availability, unravelling through aspects in the metabolism of tumor stromal cell types [100].

### 5.8. Targeting CAFs Could Create New Therapeutic Avenues in Pancreatic Cancer Therapy

Numerous research groups made great efforts in targeting CAFs for clinical benefit. Novel therapeutic approaches targeted mainly direct CAFs cells, through reprogramming of CAFs into a quiescent fibroblast, or tried to block the crosstalk between CAFs and adjacent cells. Although some of the results were quite promising, CAF targeting was confronting with many difficulties, mainly due to heterogeneity of CAF or the lack of CAF-specific markers, as previously emphasized [113].

Complete depletion of CAFs in GEMM of PDAC models by using CAF-related cell surface markers (selective depletion of α-SMA positive cells) generated surprisingly results, respectively, increased metastatic spread, enhanced intratumoral infiltration of immunosuppressive regulatory T cells, and reduced survival [169]. In contrary, Feig C. demonstrated that the depletion of FAP-expressing cells, by conditional ablation of FAP+ cells using diphtheria toxin led to an increase in anti-tumorigenic cytotoxic CD8+ T cells, slowing the pancreatic tumor growth [102,113,170].

Based on these findings, novel therapeutic strategies could be explored, addressing, on one hand, the reprogramming CAFs into quiescent fibroblasts, which could be achieved by using vitamin A metabolism/all-trans retinoic acid (ATRA), which switches both PSCs and CAFs into more quiescent fibroblast [171,172], or using vitamin D receptor or calcipotriol [173]. On the other hand, targeting interaction between CAFs and their surrounding microenvironment could be achieved by addressing the following signaling molecules:-TGFβ and interleukin signaling—blocking antibodies inhibitors;-NFkB and TNFα signaling, to reduce perlecan secretion;-Cancer cells–ECM interaction: Hedgehog signaling through IPI-926 (sonidegib and vismodegib) or hyaluronic acid through enzymes (PEGPH20) blocking antibodies inhibitors-Immunosuppression in the tumor microenvironment [113].

### 5.9. Future Directions

Considering the abovementioned aspects, we believe that further large studies are needed to explore the roles of CAFs in the development and progression of PDAC. Nevertheless, the lack of definitive markers in order to better characterize the dynamic CAF phenotype and the extreme heterogeneity of CAF and its impact on pancreatic cancer physiopathology represent research hot topics for further investigations [102]. Even less information is available about some other cancer-associated cell types.

## 6. Signaling Side: TGFβ, CD36, and CD97—Signaling in Pancreatic Cancer

The most frequent form of pancreatic cancer is PDAC, and there are multiple levels of modification in the cell environment, involving several most frequent extracellular molecules. An analysis using the Reactome Pathway Browser [174] revealed multiple pathways that appear deregulated in cancers, (Figure 2). Certain molecular key points outlined in pancreatic cancer, and mostly in PDAC, are connected to the TGFbeta pathway, including CD36.

The superfamily of TGFβ consists of more than forty members, including TGFβs, bone morphogenic proteins (BMPs), Activin, etc. [175,176]. The signaling pathway TGFβ/SMAD4 controls transduction from plasma membrane to nucleus and affects a large number of processes, such as proliferation, differentiation, apoptosis, migration, and cancer initiation and progression. TGFβ has dual actions toward tumorigenesis, exhibiting a suppressive role in early stages of tumor development, by inducing cell cycle arrest and apoptosis, while on later stages of tumor progression, cells become insensitive, and the secreted TGFβ enhances immunosuppression and promotes angiogenesis, invasion, and metastasis [175]. As the main mediator of the TGFβ pathway, SMAD4 plays a key role in switching TGFβ function on tumorigenesis. TGFβ/SMAD4 can be extensively regulated by other pathways, MAPK, PI3K/AKT, and WNT/β-catenin, forming a complex network [175,177,178,179].

If an ectopic overexpression of SMAD4 is induced in SMAD4-negative cells, this can bind the p21 promotor and enhance its transcription [180]. TGFβ is not able to induce p21 expression in pancreatic cell lines lacking SMAD4, resulting in out-of-control cell growth [181].

It was also found that TGFβ/SMAD4 may enhance p27 expression, while p27 inhibition using siRNA blocks cell growth arrest related to TGFβ/SMAD4 [182].

It was demonstrated that the TGFβ-inducible early response gene (TIEG) provoked apoptosis of pancreatic epithelial cells [183].

### 6.1. Crosstalk with Other Pathways

In the last twenty years, reports have been published on an increased number of pathways interacting with the canonical TGFβ/SMAD4. Such crosslinks appear along the whole chain of TGFβ/SMAD4 signal transduction, from the phosphorylation of SMAD2/3, formation of SMAD complex, and translocation to the nucleus.

A summary of the main interactions between the TGFβ/SMAD4 pathway and other pathways, such as MAPK (mitogen-activated protein kinase), PI3K/AKT (phosphatidylinositol-3 kinase/AKT), and WNT/β-catenin pathways, is illustrated in Figure 2.

Numerous studies proved that alteration of SMAD4 is closely associated with pancreatic cancer. In about 60% of human pancreatic cancers, the loss of heterozygosity occurs, and about 50% are presenting homozygous deletion or inactivating mutations [185,186]. The deletion of SMAD4 (both heterozygous and homozygous deletion) was initially discovered in pancreatic duct adenocarcinoma, followed by its detection in other cancers, like gastric, prostate, or colorectal cancers, albeit at reduced frequency compared to PDAC [1,2]. Another study demonstrated that SMAD4 mutation was associated with pancreatic tumor stages; the degree of inactivation was 31% in high-grade stage neoplasms (Pan IN-3), while none was found in the low-grade lesions (Pan IN-1-2) [187]. Knockout of SMAD4 by PDX1-Cre or P48 did not trigger cancer in mice [188,189], but it could facilitate tumor progression due to activation of KRASG12D [190] or inactivation of PTEN [191]. These studies suggest a tumor suppressive role of SMAD4 in progressive stages.

Of similar importance is, mainly in PDAC, the Hedgehog signaling. A schematic of the biogenesis and of the Hedgehog (Hh) “on” and “off” signaling is presented in Figure 3, which illustrates both the biogenesis of Hh and its externalization, a process involving participation of ER and Golgi (note the presence of palmitoylation and of cholesterol on the Hh molecule). The externalization is achieved by the association of Hh–Np to DISP2 and to SCUBE2, with the latter serving as an externalization factor. Without the Hh, cytosolic Gli undergo proteolytic cleavage, resulting in a form that is able to translocate to the nucleus, where it represses the transcription of target genes. Binding Hh to the cell surface receptor Patched (Ptc) stabilizes integral Gli proteins in their transcriptional activator form, thus stimulating Hh-dependent gene expression [192,193,194,195].

Hedgehog signaling has an important relationship to tumorigenesis [196,197]. Overexpression of Hh and Gli1 associates with the start of pancreatic ductal neoplasia [198,199]. In pancreatic cancer was reported a ligand-dependent activation of Hedgehog signaling, while in other cancers, genomic mutations were reported. The Sonic Hedgehog (Shh) overexpression seems to trigger the onset of pancreatic cancer [196]. Shh, along with that of Smo and Gli, is present on stromal-derived PSCs. Meanwhile, the ligands Shh and Ihh are expressed only in pancreatic tumor cells. The genetic ablation of Smo did not affect the development of PDAC tumors, suggesting the involvement of paracrine signaling [200,201]. Treating PSCs with Shh and Ihh resulted in the upregulation of the Hedgehog pathway and induced PSC proliferation. The Hedgehog signal ligands released by cancer cells induced the secretion by PCDs of cancer cells stimulating factors [202].

The involvement of both TGFβ/SMAD and of Hedgehog pathways was demonstrated in several studies regarding the oncogenesis processes [116,203], although without clear evidence of a real crosstalk between the two pathways.

### 6.2. Involvement of CD36 and CD 97 in Signaling Pathways

CD36 and CD97 interfere with complex signaling network, finetuning the cellular responses to this devastating disease. The ability of CD36 to bind different ligands provides its functional diversity. As a lipid translocase, CD36 can facilitate the transfer of lipid molecules, including LCFAs (171), ox-LDL (172), anionic phospholipids (173), and oxidized phospholipids (174). Membrane-bound CD36 is able to transfer fatty acids to the fatty acids binding protein from the cytosol and further transport them into mitochondria, thus providing energy to the cell. CD36 is able to bind other ligands, for instance, amyloid proteins, AOPPs, TSP-1, TSP-2, advanced glycation end products (AGEs), and AOPPs (175) and (176). TSP-1 is capable of binding to the CLESH domain on CD36, present on MVECs, followed by subsequent activation of Src pathway, thus influencing apoptosis of the endothelial cells. Acting as scavenger receptor, CD36 binds to other transduction proteins on the cell surface, like integrins and CD9 or CD91, mediating binding and transduction of signal. Acting as a modulator of the Toll-like receptors 4 and 6, CD36 may moderate the transduction of inflammatory signals, when meeting ox-LDL and exogenous stimuli [204] (see Figure 4).

Several studies connect CD36 and TSP-1 to the TGFβ1 signaling, linking this to pancreatic cancer [205], providing information on another molecule involved in the process, PAI-1, which is upregulated in pancreatic cancer cells. Other proofs of the involvement of CD36 and TSP-1 were provided for the decrease of protein expression of CD36 in colon cancer (with progression of the decrease from adenoma to carcinoma) and provided insights on the roles of CD36 as suppressor of the β-catenin/c-myc signaling by promotion of proteasome-dependent ubiquitination of GPC4 [206]. Such details did not appear so far for pancreatic cancer. However, for PC was reported that CD36 is required on immune cells, to allow extravasation of tumor microvesicles from premetastatic foci, while it seems that CD36 may also act as a tumor-suppressive protein, since it appears downregulated on PC cells [18,66].

The presence of CD97 in cancers like pancreatic, gastric esophageal, or thyroid suggests that its expression might be a common characteristic of such tumors. The results were confirmed on a number of other carcinomas [81]. The potential interaction of CD97 with CD55 situated in the extracellular matrix and its importance for tumor invasion are supported by results regarding enhanced CD97 in scattered tumor cells present in the invasion front. Such cells in the invasion front displayed a modification of expression or function of the adhesive E-cadherin–catenin complex [207,208]. Beta-catenin, which is usually expressed on the cell membrane, presented abnormal accumulation in the cell nuclei, with the process having a major role in acquiring an invasive phenotype [209]. CD97 has a distinct pattern of expression in the pancreas, where it is present in higher amounts than in other tumoral and peritumoral tissues, where it usually shows a very low presence [81,210,211]. In the pancreatic ducts, where most pancreatic adenocarcinoma originates, the pancreatic progenitor cells are CD97+. The pattern of CD97 expression in normal tissue is parallel with that of Ep-CAM. Epithelial growth by budding from ductal cells induces the upregulation of Ep-CAM, while the differentiation into endocrine cells generates a downregulation of the same molecule. This shows characteristics of a molecule acting in a morpho-regulatory and differentiation-dependent manner. However, it is still left to speculation whether CD97 is acting by a similar pattern on pancreatic progenitor cells or if it plays a more general role in the process of differentiation.

Exploration of signaling pathways, both of the TGF-beta dependent and Hedgehog and of the crosstalk with CD36 and CD97, or of these with other intracellular signaling molecules offers potential solutions for stratification of pancreatic cancers and optimization of therapies, based on particular aspects of each subset.

## 7. Favoring Quiescence (Cell Dormancy)—A Valid Therapeutic Strategy in Pancreatic Cancer?

Cell quiescence can be defined as a cellular state/phenotype that cells can enter or exit, acting as a switch and allowing tuning (during health or disease) of regenerative/proliferative mechanisms. Quiescence is a versatile resource cells actually exploit, be they hematopoietic stem cells [212,213], satellite cells in skeletal muscle [214], or subpopulations of neoplastic cells [215].

In this case, as a “dormant cellular state, which dictates the distinct tumorigenic aggressiveness” [216], beside mechanisms operating in normal cells, there are probably complementary mechanisms that are less-explored and possible double-edged sword.

Recently, systematic information about quiescence in different cell types has been constantly accumulating, e.g., neurons [216], hair follicle stem cells [217], muscle stem cells [217], vessels (“During quiescence, the angiogenic switch is ‘off’”) [218], vessels during diabetes [219], pancreatic stem cells [109], beta-cells during diabetes [220], mesenchymal stem cells [221], or pancreatic progenitor-like acinar cells [222].

It was found reasonable to inquire about the possibility to search (and find) factor(s) responsible for quiescence or senescence of pancreatic beta-cells (i.e., “the failure of cyclin-dependent kinases (CDKs) and cyclins to access the nucleus”), in order to reverse the clinical status by favoring exit quiescence [220].

On the opposite, one may put the question if to induce a type of quiescence similar with that observed in normal cells could be, as well, a reasonable therapeutic strategy in pancreatic cancer.

During neoplasia, a series of dis-equilibria concerning complex intricate mechanisms were revealed, and each of them was, at its turn (more or less), examined in terms of neoplastic (or metastatic) progression.

For tumor cells, the balance between quiescence and activation of different cell subtypes may favor tumor development or therapy resistance. Poles apart, from a therapeutical point of view, altering this equilibrium (favoring a quiescence similar with that operating in stem cells or neurons) could be, a priori, an effective strategy.

In another type of cancer, human mesenchymal stem cells (hMSC) were shown contribute to quiescence and therapy resistance of persistent acute myeloid leukemia (AML cells) [223]. In this study, quiescence is related to therapy resistance, and thus with an unwanted result. However, what facts are currently documented for pancreatic cancer that explicitly explore the relationship between quiescence and therapeutically outcome? As a pure speculation, inducing quiescence obviously does not cure the disease, but, again, a priori, if effective, manipulating quiescence may gain time in a disease with a dramatic evolution in a short time.

Different markers associated with quiescence were identified, in different normal and tumoral tissues. Leucine-rich repeats and immunoglobulin-like domains 1 (Lrig1) regulate cultured neural stem cell quiescence. The marker was identified in multiple organs in mice [224], but there is only a paper examining Lrig1 in pancreatic cancer [225].

Quiescence and long-term maintenance of different cell populations’ subtypes were documented in different (including pancreatic) cancer types [109,215,226]. Table 3 presents some examples of facts/or proposed mechanisms contributing to the understanding of quiescence of different pancreatic cancer cell types.

Facts (experimental findings) presented in this paper suggest that a long-term induced quiescence of specific cell types in the pancreas may turn down the evolution of pancreatic cancer, through mechanisms that are currently still controversial and speculative. However, the hypothesis deserves a more thorough examination, taking into account the slow progress in improving prognosis in this disease, despite the extensive new approaches.

The relative proportion of cell types required to block or delay progression remains to be established. One may speculate that it is possible that one single cell type reconverted to quiescence to divert evolution to a better stage, but it is also possible to achieve quiescence only when several cellular types act together in a coordinated manner.

## 8. Conclusions

Considering the abovementioned aspects, further large studies are needed to explore the roles and behavior of different cancer-associated cells in the development and progression of PDAC. Nevertheless, the lack of definitive markers in order to accurately characterize the dynamics of cancer-associated cells’ phenotypes, the extreme heterogeneity of these cells, and its impact on pancreatic cancer pathophysiology represent hot research topics for further investigations.

Exploration of signaling pathways, both of the TGF-beta dependent and of the crosstalk with CD36 and CD97, or with other intracellular signaling molecules, offers potential hints for identification and stratification of pancreatic cancer cell subtypes, cell cooperation in tumor microenvironment, and optimization of therapies, based on particular aspects of each subset.

Reversal or induction of a quiescence state in cancer-associated target cells deserves further exploration in pancreatic cancer, and there is an acute need of new biomarkers able to identify not only specific cell subtypes, but also states of specific cellular subtypes.

Quiescence is a cellular state that is long-lasting in normal cells, and understanding how to induce such a state in aggressively proliferating pancreatic neoplasia could, a priori, provide the means to delay evolution and improve outcome.

At least two recent observations support this view:(a)“Therefore, reversal of activated fibroblasts to the quiescence state is an important area of investigation that may help the therapeutic management of a number of diseases including pancreatic cancer” [227].(b)“Thus, targeting the CAFs at this stage with molecules that can revert the back to “quiescent” state can be considered an attractive therapeutic strategy, as this will disrupt the tumor–stroma crosstalk and inhibit the tumor growth and progression” [228].

## Figures and Tables

**Figure 1 ijms-21-05656-f001:**
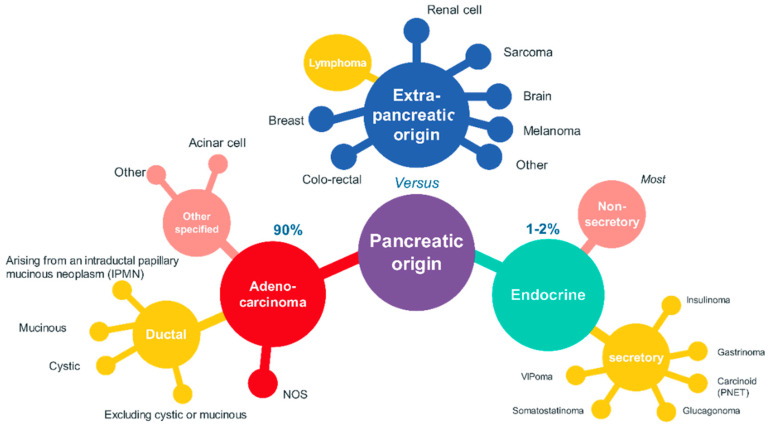
Histological types of pancreatic cancer, based on References [15,16].

**Figure 2 ijms-21-05656-f002:**
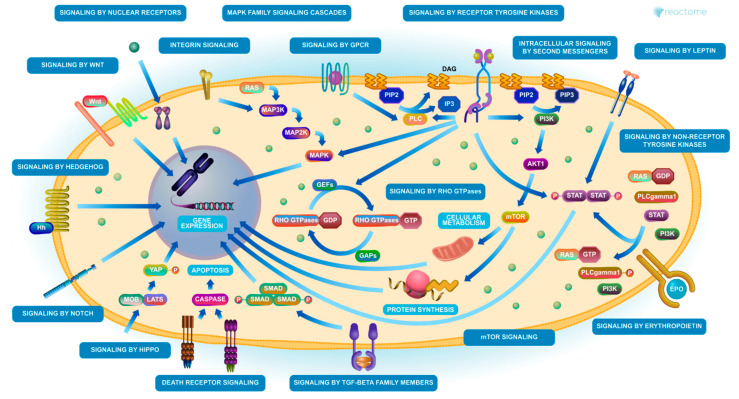
Overall signaling cascades, including the TGF-beta SMAD cascade (reproduced with permission of Reactome Pathway, Fabregat et al. [184]).

**Figure 3 ijms-21-05656-f003:**
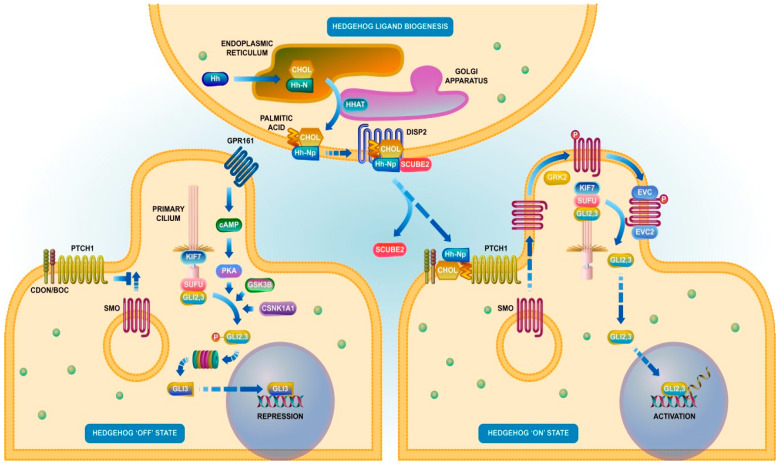
Schematic of biogenesis and of the “off”—“on” Hedgehog signaling. Reproduced with permission from Reactome (https://reactome.org/PathwayBrowser/#/R-HSA-5358351).

**Figure 4 ijms-21-05656-f004:**
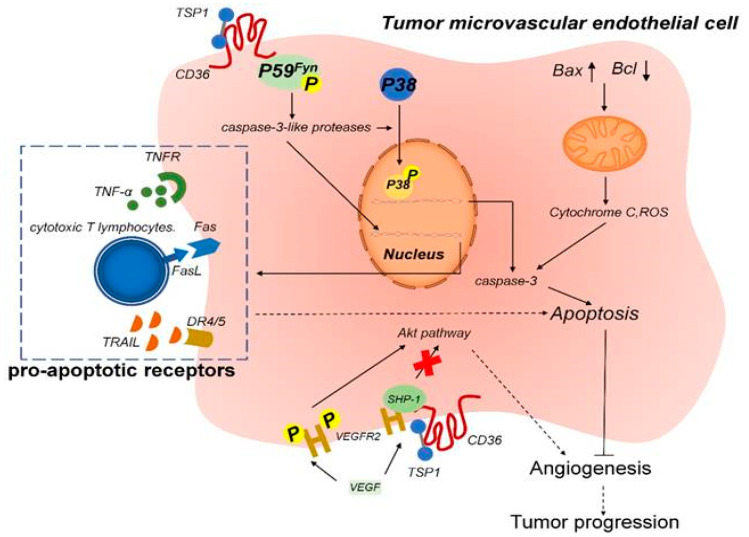
TSP-1-CD36 signaling is inducing apoptosis of tumor associated epithelial cells. As a result of TSP-1 binding to CD36 on microvascular endothelial cells, phosphorylation and, therefore, activation of P59fyn (cytoplasmic protein kinase) occur. This in turn stimulates caspase-like proteases, which induce the phosphorylation of MAPK. Nuclear translocation of MAPK generates increased expression of caspase-3 and of proapoptotic receptors, leading to apoptosis. Mitochondrial damage leads to the release of reactive oxygen species and of cytochrome C, which are also triggers of apoptosis. Moreover, the binding of THC-1 to CD36 induces the recruitment of SHP-1 to the VEGFR2 complex, followed by VEGFR2 dephosphorylation and inhibition of the VEGF pathway and leading to anti-angiogenesis. (Reproduced with permission from Reference [53].)

**Table 1 ijms-21-05656-t001:** Current mechanisms that can convert normal fibroblasts into cancer-associated fibroblast.

Process/Changes	Activated Molecules	References
Activation Ligands	transforming growth factor-β (TGFβ), platelet-derived growth factors (PDGFs), epidermal growth factors (EGFs), fibroblast growth factors (FGFs), bone morphogenic proteins (BMPs), Sonic Hedgehog (Shh)	[103]
Contact Signals	notch signaling	[118]
Inflammatory Modulators	IL-1 →NFkB and IL-6 →STAT transcription factors → JAK–STAT signaling → contractile cytoskeleton and histone acetylation	[119,120]
Physical Changes in EC	stiffness and composition	[121,122,123]
Physiological Stress	heat-shock factors	[124]
Genomic Stress/DNA Damage—Chemo-/Radiotherapy	ROS	[106,125]

**Table 2 ijms-21-05656-t002:** Cancer-associated fibroblasts regulate cancer progression through a dynamic secretome.

Molecules Released by CAFs	Modified Processes that Orchestrate Tumor Development and Immune Evasion	References
VEGFs, PDGFs, HGF, IL-8, SDF-1	*enhanced angiogenesis*	[140,141]
IL-6, IL-1	*enhanced inflammation*	[142]
IL-6, TGFβ, SDF-1, HGF, lysophosphatidylcholines-LPCs	*enhanced cell proliferation*	[143,144,145]
TGFβ, COX-2/PGE2	*enhanced motility*	[146]
IL-6, CXCL 12	*macrophage switch and immune evasion*	[147]
HIF1a, lactate dehydrogenase A	*altered metabolism*	[148]
differentiation factors, activin A, FGF2	*altered cell fate*	[149]
TGFβ, HGF, FGFs, NGF, IGF	*enhanced secretion of cytokines and growth factors*	[150,151]
Fibronectin, collagen 1, tenascin C, osteopontin—MMPs	*deposition and remodeling of ECM*	[141,152,153,154]

ECM = extracellular matrix.

**Table 3 ijms-21-05656-t003:** Mechanisms involved in quiescence of different pancreatic cancer cell types.

Cell Type	Positioning (Tumor or Associated)	Facts (or Proposed Mechanisms)	References
Stromal	“Pancreatic cancer-associated fibroblasts”	“Gold nanoparticle transforms activated cancer-associated fibroblasts to quiescence by enhancing lipid synthesis and lipid utilization”.	[227]
		“Anticancer compound Minnelide revealed deregulation of the TGFb signalling pathway in CAF,	[228]
		resulting in an apparent reversal of their activated state to a quiescent, nonproliferative state”.	[229]
		“This heterogeneity explains why one type of CAF is found to support cancer invasiveness and metastases while another type does not”.	
	“Pancreatic cancer-associated adipocyte”	“Cancer-associated adipocytes exhibit distinct phenotypes and facilitate tumor progression in pancreatic cancer”, but quiescence was not examined.	[230]
	Pancreatic tumor-associated macrophages	“The expression of homeobox protein VentX, a master regulator of macrophage plasticity, is significantly decreased in the PDA-TAMs”.	[231]
	Stellate cells	“Up-regulation of Ppar-γ which is associated with quiescence”.	[214]
		“In the healthy pancreas, PSCs are in the quiescent state and retain vitamin A-containing lipid droplets”.	[232,233]
		“PSC, quiescent in the healthy pancreas. During pancreatic injury, PSC develop a myofibroblast phenotype expressing αSMA1, actively proliferate and migrate. Activation of PSC is promoted by TGFβ, HGF, FGF, EGF, and sHH”.	[234]
		“p53 activation by Nutlin-3a induces profound transcriptional changes, which reprogram activated PSCs to quiescence”.	
Cancer stem cell		Quiescent stem cells are characterized by high chemo-resistance, clonogenic ability, and metastatic potential.	[235]
Pancreatic progenitor-like cell		“Dclk1+ and Stmn+ cells are long-lived, largely quiescent, and lack proliferation under resting conditions”.	[222]

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
