# Peer review of "CD36 and CD97 in Pancreatic Cancer versus Other Malignancies"

_ijms, 2020, doi:10.3390/ijms21165656_

Round 1
Reviewer 1 Report
This is an interesting review on the role of CD36 and CD97 as potential markers of “quiescence” in pancreatic cancer. Overall, the paper is well designed and written, containing also pertinent figures.
The paragraph on epithelial-mesenchymal transition should be better discussed and presented (suggested references PMID: 25989272, 30669452, 24062980), with associations with TGF-beta signaling (not only SMAD4–linked: there are also other genes, belonging to this pathway, which are important in pancreatic cancer).
Hedgehog signalling pathway is only marginally discussed. Please improve the discussion of this importan topic (suggested references PMID: 31979397, 29521941).
A significant part on the techniques to investigate the presence of CD36 and CD97 in tissues should be better presented, with a clear focus on immunohistochemistry, transcriptomics,…
Reviewer 2 Report
The review article entitled: "Facts about CD36 and CD97 in pancreatic cancer" by Cristiana Tanase et al. is overall well writen and well introduced. The manuscript focus on increasing crucial roles of CD36 and CD97 in the microenvironment of pancreatic cancer that may lead to its aggressiveness and development. Furthermore, authors provide the rational to design novel therapeutic approaches against pancreatic cancer based on this markers. Please, find below my recommendations to improve the quality of the review.
1.- In figure 1, I suggest to show only the schema of pancreatic origin due the extra-pancreatic lacks several other tumor types like brain tumors or other squamous histological subtypes among others.
2.- Please define "FA" in line 84.
3.- Please include a reference to Pang et al. Food Funct. 2019 and the role of querceptin in CD36.
4.- Please discuss further the fact that CD36 is found in significantly lower in cancer cells than in corresponding non-tumor normal tissues, and the lower the CD36 level in the stroma, the more aggressive the tumor; however, it contributes to progression and metastasis, activation of cancer stem cells, epithelial to mesenchymal transition, chemoresistance and it is considered an unfavorable prognostic factor in PDAC.
5.- Please include more data about clinical prognostic studies of PDAC cited in line 135.
6.- Unify: FA or FAs or fatty acid (line 84 vs line 155 vs line 160).
7.- et al is an abbreviation, then must be replaced by et al. with a point.
8.- Attention on citation of line 152 and 156 and 160.
9.- PDAC of line 237 has already been defined previosly.
10.- (Line 288), Then CD97 could only differentiate between pancreatitis and pancreatic duct cell carcinoma well differentiated?
11.- For multiple mechanisms of CAFs activation (line 407), please design a explanatory table.
12.- Please design a table for the cathegorized secretome including the references (line 464).
Reviewer 3 Report
The review titled “Facts about CD36 and CD97 in pancreatic cancer” by Tanase et al is interesting and is within the scope of Journal. It needs revision before it can be considered for publication.
- Suggestion: Title needs to be reworded. Also, just CD36 and CD97 in the titles does not reflect to a majority of audience what will be the content of the paper.
- “FA” has not been defined in the text.
- “2.2 CD36 promotes tumor metastasis”
This paragraph looks like describing the role of CD36 in different tumor metastasis and not just limited to Pancreatic cancer. The paragraph is very generic and does not specify which cancer type is being referred while describing the case. The readers are repeatedly required to check the references to understand in which cancer type the relevant information i
Has been obtained. Please clarify the cancer type in each case in the text.
- In different places in the review, why PMID is referred than a citation?
- “2.4 CD36 can regulate chemoresistance and radioresistance”seems specific to pancreatic cancer. Is it not? If yes then include pancreatic cancer in the title. The subheadings are confusing. Does not clarify whether the authors talk about the CD36 and CD97 in different cancer or the information is specific to pancreatic cancer.
- Why to examine concomitant expression of CD36 and CD97s? (Why bother about CD36 and CD97 in pancreatic cancer?) The question asked is not well answered in the described paragraph.
- The linkers or transition between CD36 and CD97 to fibroblasts in Pancreatic cancer is missing.
- The review lacks a flow of idea. It appears as a collection of information about CD36 and CD97 in any cancer and then any molecule related with fibroblasts.
Round 2
Reviewer 2 Report
Thanks for the appropiate and amended version of the review. Congratulations.
Reviewer 3 Report
The manuscript has improved significantly. It can be accepted in the current form.